# An International Retrospective Observational Study of Liver Functional Deterioration after Repeat Liver Resection for Patients with Hepatocellular Carcinoma

**DOI:** 10.3390/cancers14112598

**Published:** 2022-05-24

**Authors:** Zenichi Morise, Luca Aldrighetti, Giulio Belli, Francesca Ratti, Tan To Cheung, Chung Mau Lo, Shogo Tanaka, Shoji Kubo, Yukiyasu Okamura, Katsuhiko Uesaka, Kazuteru Monden, Hiroshi Sadamori, Kazuki Hashida, Kazuyuki Kawamoto, Naoto Gotohda, KuoHsin Chen, Akishige Kanazawa, Yutaka Takeda, Yoshiaki Ohmura, Masaki Ueno, Toshiro Ogura, Kyung Suk Suh, Yutaro Kato, Atsushi Sugioka, Andrea Belli, Hiroyuki Nitta, Masafumi Yasunaga, Daniel Cherqui, Nasser Abdul Halim, Alexis Laurent, Hironori Kaneko, Yuichiro Otsuka, Ki Hun Kim, Hwui-Dong Cho, Charles Chung-Wei Lin, Yusuke Ome, Yasuji Seyama, Roberto I. Troisi, Giammauro Berardi, Fernando Rotellar, Gregory C. Wilson, David A. Geller, Olivier Soubrane, Tomoaki Yoh, Takashi Kaizu, Yusuke Kumamoto, Ho-Seong Han, Ela Ekmekcigil, Ibrahim Dagher, David Fuks, Brice Gayet, Joseph F. Buell, Ruben Ciria, Javier Briceno, Nicholas O’Rourke, Joel Lewin, Bjorn Edwin, Masahiro Shinoda, Yuta Abe, Mohammed Abu Hilal, Mohammad Alzoubi, Minoru Tanabe, Go Wakabayashi

**Affiliations:** 1Department of General Surgery, Fujita Health University School of Medicine Okazaki Medical Center, Okazaki 444-0827, Japan; 2Hepatobiliary Division in Department of Surgery, San Raffaele Hospital, 20132 Milano, Italy; aldrighetti.luca@hsr.it (L.A.); ratti.francesca@hsr.it (F.R.); 3Department of General and HPB Surgery, Loreto Nuovo Hospital, 80127 Naples, Italy; chirurgia.loretonuovo@tin.it; 4Division of HBP and Liver Transplant, University of Hong Kong Queen Mary Hospital, Hong Kong, China; tantocheung@hotmail.com (T.T.C.); chungmlo@hku.hk (C.M.L.); 5Department of Hepato-Biliary-Pancreatic Surgery, Osaka City University Graduate School of Medicine, Osaka 545-8586, Japan; m8827074@msic.med.osaka-cu.ac.jp (S.T.); m7696493@msic.med.osaka-cu.ac.jp (S.K.); 6Division of Hepato-Biliary-Pancreatic Surgery, Shizuoka Cancer Center Hospital, Sunto, Shizuoka 411-8777, Japan; yu.okamura@scchr.jp (Y.O.); k.uesaka@scchr.jp (K.U.); 7Departments of Surgery, Fukuyama City Hospital, Fukuyama 721-8511, Japan; monden0319@yahoo.co.jp (K.M.); shimin-byouin@city.fukuyama.hiroshima.jp (H.S.); 8Department of Surgery, Kurashiki Central Hospital, Kurashiki 710-8602, Japan; kh14813@kchnet.or.jp (K.H.); kk7159@kchnet.or.jp (K.K.); 9Division of Hepatobiliary and Pancreatic Surgery, National Cancer Center Hospital East, Kashiwa 277-8577, Japan; ngotohda@east.ncc.go.jp; 10Division of General Surgery, Department of Surgery, Far-Eastern Memorial Hospital, New Taipei City 220, Taiwan; chen.kuohsin@gmail.com; 11Department of Electrical Engineering, Yuan Ze University, Taoyuan City 320, Taiwan; 12Department of Hepato-Biliary-Pancreatic Surgery, Osaka City General Hospital, Osaka 534-0021, Japan; kanazawaaki@mac.com; 13Department of Surgery, Kansai Rosai Hospital, Amagasaki 660-8511, Japan; takeda-yutaka@kansaih.johas.go.jp (Y.T.); ohmura-yoshiaki@kansaih.johas.go.jp (Y.O.); 14Second Department of Surgery, Wakayama Medical University, Wakayama 641-8509, Japan; ma@wakayama-med.ac.jp; 15Department of Hepatobiliary and Pancreatic Surgery, Graduate School of Medicine, Tokyo Medical and Dental University, Tokyo 113-8510, Japan; ogumsrg@tmd.ac.jp (T.O.); tana.msrg@tmd.ac.jp (M.T.); 16Department of Hepatobiliary and Pancreatic Surgery, Seoul National University Hospital, Seoul 03080, Korea; kssuh2000@gmail.com; 17Department of Gastrointestinal Surgery, Fujita Health University School of Medicine, Toyoake 470-1192, Japan; y-kato@fujita-hu.ac.jp (Y.K.); sugioka@fujita-hu.ac.jp (A.S.); 18Department of Abdominal Surgical Oncology, Fondazione G.Pascale-IRCCS, National Cancer Institute of Naples, 80131 Napoli, Italy; a.belli@istitutotumori.na.it; 19Department of Surgery, Iwate Medical University, Morioka 028-3695, Japan; hnitta@iwate-med.ac.jp; 20Department of Surgery, Kurume University School of Medicine, Kurume 830-0011, Japan; m-yasunaga@saiseikai-futsukaichi.org; 21Paul Brousse Hospital, 94800 Villejuif, France; daniel.cherqui@aphp.fr (D.C.); nasserah@clalit.org.il (N.A.H.); 22Paris-Sud University, 91190 Gif-sur-Yvette, France; alexis.laurent@aphp.fr; 23Division of General and Gastroenterological Surgery, Department of Surgery, Toho University Faculty of Medicine, Tokyo 143-8540, Japan; hironori@med.toho-u.ac.jp (H.K.); yotsuka@med.toho-u.ac.jp (Y.O.); 24Division of Hepatobiliary Surgery and Liver Transplantation, Department of Surgery, Ulsan University and Asan Medical Center, Seoul 05505, Korea; khkim620@amc.seoul.kr (K.H.K.); hwuidongcho@gmail.com (H.-D.C.); 25Department of Surgery and Surgical Oncology, Koo Foundation Sun Yat-Sen Cancer Center, Taipei 112, Taiwan; charleslin@ircadtaiwan.com.tw; 26IRCAD-AITS, Changhua 505, Taiwan; 27Department of Surgery, Tokyo Metropolitan Cancer and Infectious Diseases Center Komagome Hospital, Tokyo 113-8677, Japan; yusuke_omen@yahoo.co.jp (Y.O.); seyamaysur-tky@umin.ac.jp (Y.S.); 28Department of Clinical Medicine and Surgery, University of Naples Federico II, 80138 Napoli, Italy; roberto.troisi@unina.it; 29General Hepato-Biliary and Liver Transplantation Surgery, Ghent University Hospital Medical School, 9000 Gent, Belgium; gberardi1@gmail.com; 30Hepato-Bilio-Pancreatic Unit of Clinica Universitaria de Navarra, 31008 Pamplona, Spain; frotellar@gmail.com; 31Department of Surgery, University of Pittsburgh, Pittsburgh, PA 15213, USA; wilsongc@upmc.edu (G.C.W.); gellerda@upmc.edu (D.A.G.); 32Department of HPB Surgery and Liver Transplant, Beaujon Hospital, Clichy 92110, France; olivier.soubrane@gmail.com (O.S.); tomyoh@kuhp.kyoto-u.ac.jp (T.Y.); 33Department of Surgery, Kitasato University School of Medicine, Sagamihara 252-0374, Japan; t-kaizu@kitasato-u.ac.jp (T.K.); kumamoto@kitasato-u.ac.jp (Y.K.); 34Seoul National University College of Medicine, Bundang Hospital, Seongnam-si 13620, Korea; hanhs@snubh.org (H.-S.H.); eekmekcigil@gmail.com (E.E.); 35Antoine Beclere Hospital, 92140 Clamart, France; ibrahim.dagher@aphp.fr; 36Department of Digestive Diseases, Institute Mutualiste Montsouris, University of Paris Descartes, 75014 Paris, France; davidfuks80@gmail.com (D.F.); brice.gayet@imm.fr (B.G.); 37Tulane Transplant Abdominal Institute, Tulane University, New Orleans, LA 70112, USA; joseph.buell@hcahealthcare.com; 38Unit of Hepatobiliary Surgery and Liver Transplantation, University Hospital Reina Sofia, 30003 Murcia, Spain; rubenciria@gmail.com (R.C.); javibriceno@hotmail.com (J.B.); 39Department of General Surgery and HPB Surgery, Royal Brisbane Hospital, The University of Queensland, St Lucia, QLD 4072, Australia; orourke.nick@gmail.com (N.O.); joel.lewin@uqconnect.edu.au (J.L.); 40Department of Hepatopancreatobiliary Surgery, Oslo University Hospital-Rikshospitalet, 0372 Oslo, Norway; bjoedw@ous-hf.no; 41Department of Surgery, Keio University School of Medicine, Tokyo 160-8582, Japan; masa02114@yahoo.co.jp (M.S.); abey3666@gmail.com (Y.A.); 42Istituto Ospedaliero—Fondazione Poliambulanza, 25124 Brescia, BS, Italy; abuhilal9@gmail.com; 43University Hospital Southampton, Hampshire SO16 6YD, UK; mhm0001900@yahoo.com; 44General Surgery Department, The University of Jordan, Amman 11972, Jordan; 45Department of Surgery, Ageo Central General Hospital, Ageo 362-8588, Japan; go324@mac.com

**Keywords:** laparoscopic liver resection, repeat liver resection, liver function, liver functional deterioration, overall survival

## Abstract

**Simple Summary:**

For 657 cases of segment or less repeat liver resection with results of plasma albumin and bilirubin levels and platelet counts before and 3 months after surgery, the indicators were compared before and after surgery. There were 268 open repeat after open and 224 cases laparoscopic repeat after laparoscopic liver resection. The background factors and liver functional indicators before and after surgery, and the changes were compared between both groups. Plasma levels of albumin (*p* = 0.006) and total bilirubin (*p* = 0.01) were decreased, and ALBI score (*p* = 0.001) indicated worse liver function after surgery. Though laparoscopic group had poorer performance status and liver function, changes of the values and overall survivals were similar between both groups. Plasma levels of albumin and bilirubin and ALBI score could be the liver functional indicators for liver functional deterioration after liver resection. The laparoscopic group with poorer conditions showed a similar deterioration of liver function and overall survival to the open group.

**Abstract:**

Whether albumin and bilirubin levels, platelet counts, ALBI, and ALPlat scores could be useful for the assessment of permanent liver functional deterioration after repeat liver resection was examined, and the deterioration after laparoscopic procedure was evaluated. For 657 patients with liver resection of segment or less in whom results of plasma albumin and bilirubin levels and platelet counts before and 3 months after surgery could be retrieved, liver functional indicators were compared before and after surgery. There were 268 patients who underwent open repeat after previous open liver resection, and 224 patients who underwent laparoscopic repeat after laparoscopic liver resection. The background factors, liver functional indicators before and after surgery and their changes were compared between both groups. Plasma levels of albumin (*p* = 0.006) and total bilirubin (*p* = 0.01) were decreased, and ALBI score (*p* = 0.001) indicated worse liver function after surgery. Laparoscopic group had poorer preoperative performance status and liver function. Changes of liver functional values before and after surgery and overall survivals were similar between laparoscopic and open groups. Plasma levels of albumin and bilirubin and ALBI score could be the indicators for permanent liver functional deterioration after liver resection. Laparoscopic group with poorer conditions showed the similar deterioration of liver function and overall survivals to open group.

## 1. Introduction

The treatment options for hepatocellular carcinoma (HCC) are liver resection (LR) [1], liver transplantation [2], transarterial chemoembolization, local ablation therapy [3], and currently emerging systemic (immune-) chemotherapy using kinase inhibitors and immune checkpoint inhibitor [4,5]. Although some treatments provide the hope for a cure of the current HCC [3,6,7,8], most patients of HCC with underlying chronic liver disease (CLD) are developing metachronous multicentric HCCs from its preneoplastic background. When considering treatments for the patients, not only the oncological therapeutic effects to the current tumor, but also the post-treatment residual liver function for the future HCC treatments should be taken into account. The strategy of combination therapy during the long treatment history of HCC patients, depending on each patient’s tumor condition and liver function at each time, is needed [9,10]. Although the strategy should be planned with liver functional assessments of the deterioration after treatments, there is currently no good tool for the assessment.

We (ILLS-Tokyo collaborator group) conducted international multi-institutional propensity score-based studies for laparoscopic repeat LR (LRLR) with patients with HCC, comparing to open repeat LR (ORLR) [11,12]. In the study [11], the overall survival curves after LRLR and ORLR were clearly separated with the better tendency in LRLR (not significant with *p*-value of 0.086), although the disease-free survival curves were identical and overlapped. We speculated that overall survival after LRLR was better since less liver functional damage of LRLR [13] made the repeat treatments more accessible and the number of deceased patients due to liver insufficiency decreased.

Recently, ALBI score [14,15] calculated with plasma albumin and total bilirubin levels and ALPlat score [16] calculated with plasma albumin level and blood platelet counts were proposed as the indicators of liver functional reserve for the preoperative evaluation of LR. In this study, we examined whether plasma albumin level, total bilirubin level, blood platelet counts, ALBI score, and ALPlat score could be useful as liver functional indicators for the assessment of permanently settled liver functional deterioration 3 months after repeat LR (RLR) and, using the indicators, evaluated that the extent of liver functional deterioration after LRLR compared to ORLR.

## 2. Methods

### 2.1. Participating Centers and Registered Patients

The present study involved 42 high-volume liver surgery centers around the world that provided data from patients who underwent RLR for HCC between January 2007 and December 2017. Institutional Review Board (IRB) approval was obtained from the coordinating center, with a data transfer agreement and IRB approval having been provided by all centers.

The centers registered 1582 patients, including 934 and 648 treated by ORLR and LRLR. Each case was discussed under a multidisciplinary setting in each center, and each patient provided informed consent for the procedure. The detail of registered patients’ number from each center in original patient group was described in a previous study [10].

This study conformed to the ethical guidelines of Declaration of Helsinki and was retrospective in nature. Approval from the ethics committee of each institution was obtained (HM20-094 for primary investigator’s institution, FHU).

### 2.2. Selection of Patients and Data Collection

For 1582 registered patients, the results of usual laboratory blood examination were examined. A total of 875 patients, in whom the results of plasma albumin level, total bilirubin level, and blood platelet counts before and 3 months after surgery could be retrieved, were extracted. Background factors of the patients with ORLR or LRLR are described in Table 1. Then, 657 patients, who underwent segment or less resection, were selected for the first study searching indicators for liver functional change 3 months after RLR in order to eliminate the impact of decreased liver volume after LR.

The following data were obtained as background factors: patient characteristics (age, sex, body mass index (BMI), and preoperative performance status (PS)); indicators of preoperative liver function (presence of liver fibrosis, plasma total bilirubin level (mg/dL), plasma albumin level (g/dL), blood platelet count (/microL), Child–Pugh score)); tumor characteristics (number, size (mm), and location (anterolateral or posterosuperior segments)); surgical procedures (ORLR or LRLR) and the previous LR procedure (open or laparoscopic).

In addition, the results 3 months after RLR of plasma albumin level, total bilirubin level, and blood platelet counts were obtained.

### 2.3. Analysis of the Indicators of Liver Function before and 3 Months after RLR

The results before and 3 months after RLR of plasma albumin level, total bilirubin level, and blood platelet counts were compared in the selected 657 patients. Furthermore, calculated ALBI scores [14,15] and ALPlat scores [16] before and after RLR were compared (Table 2). 

### 2.4. Comparison between the Patients Who Underwent ORLR after Previous Open LR (OO group) and LRLR after Previous Laparoscopic LR (LL Group): Background Factors, Indicators for Liver Function before RLR, Their Changes after RLR, and Overall Survival after RLR

There were 268 patients who underwent ORLR after previous open LR (OO group) and 224 patients who underwent LRLR after previous laparoscopic LR (LL group) among selected 657 patients with segment or less RLR. Selected patients’ numbers for the final analysis, comparing ORLR and LRLR in the present study, from each center are in the description of Table 3

The factors listed before (background factors, indicators for liver function, ALBI score, and ALPlat score) RLR; plasma albumin level, total bilirubin level, blood platelet counts, ALBI score, and ALPlat score 3 months after RLR were compared between LL and OO groups.

Changes of the values before and after RLR in albumin, bilirubin, platelet, ALBI score, and ALPlat score were compared between LL and OO groups.

Overall survival after RLR was compared between LL and OO groups.

### 2.5. Statistical Analyses

Data are expressed as mean ± standard deviation or as the number of patients. Between-group differences in categorical variables were analyzed by Pearson’s Chi-squared test or Fisher’s exact test with Yates correction, as appropriate. Between group differences in continuous parametric variables were analyzed by un-paired Student’s *t*-test or ANOVA, and between-group differences in continuous non-parametric variables were analyzed by Mann–Whitney or Kruskal–Wallis test. Survival was plotted by the Kaplan–Meier method, and between-group differences were analyzed by log-rank test. Statistical analyses were performed with the use of SPSS Statistics 25 (IBM Corp., Armonk, NY, USA) or R 3.3.4 (R Foundation for Statistical Computing, Vienna, Austria). *p* < 0.05 was considered statistically significant.

## 3. Results

### 3.1. Analyses of the Indicators for Liver Function before and 3 Months after RLR

Plasma levels of albumin (4.04 ± 0.45 vs. 3.97 ± 0.53 g/dL, *p* = 0.006) was significantly decreased and total bilirubin (0.76 ± 0.33 vs. 0.81 ± 0.40 mg/dL, *p* = 0.01) was significantly increased 3 months after RLR compared to those values before RLR. The difference in blood platelet counts was not significant (14.07 ± 5.02 vs. 14.12 ± 5.20 × 10^4^/microL, *p* = 0.862). Consequently, ALBI score (−2.73 ± 0.40 vs. −2.65 ± 0.48, *p* = 0.001) indicated significantly worse liver function 3 months after RLR, but not ALPlat score (504.49 ± 70.46 vs. 498.24 ± 77.05, *p* = 0.125).

### 3.2. Comparison between OO Group and LL Group: Background Factors, Indicators for Liver Function before RLR, and Those, Their Changes, and Overall Survivals after RLR

There was significantly higher BMI and poorer PS in the LL group. The LL group had significantly higher incidence of liver fibrosis and Child–Pugh score before RLR, although there were no significant differences between OO and LL groups in tumor-related factors, such as tumor number, size, and location. In addition, there were significant differences before and also after RLR in plasma level of albumin (OO vs. LL before RLR: 4.09 ± 0.39 vs. 3.94 ± 0.49 g/dL, *p* < 0.001; OO vs. LL after RLR: 4.03 ± 0.47 vs. 3.89 ± 0.55 g/dL, *p* = 0.003), blood platelet count (14.58 ± 4.89 vs. 13.57 ± 5.41 × 10^4^/microL, *p* = 0.031; 14.77 ± 5.09 vs. 13.68 ± 5.56 × 10^4^/microL, *p* = 0.025), ALBI score (−2.78 ± 0.34 vs. −2.65 ± 0.46, *p* < 0.001; −2.71 ± 0.42 vs. −2.59 ± 0.52, *p* = 0.003), and ALPlat score (514.32 ± 61.09 vs. 490.43 ± 79.95, *p* < 0.001; 510.10 ± 69.08 vs. 486.70 ± 83.60, *p* = 0.001) between LL vs. OO groups. (Table 3)

All the changes of values before and after RLR in albumin, bilirubin, platelet, ALBI score, and ALPlat score were similar without significant differences between LL and OO groups. (Table 4)

There was no significant difference in overall survival after RLR between LL and OO groups. (Figure 1, *p* = 0.576).

OO patients were registered from Clinica Universitaria de Navarra = 2 (number of patients), Wakayama Medical University Hospital = 13, Osaka City University = 30, Queen Mary Hospital = 34, Shizuoka Cancer Center = 47, University of Pittsburgh = 2, University Hospital Reina Sofia = 1, Kitazato University = 7, Komagome Hospital = 5, Osaka City General Hospital = 5, Kurume University = 18, Kurashiki Central Hospital = 18, National Cancer Center Hospital East = 37, Kansai Rosai Hospital = 4, Tokyo Medical and Dental University = 22, Toho University = 6, Fujita Health University Hospital = 4, Keio University = 3 and LL from Seoul National University Bundang Hospital = 2, Clinica Universitaria de Navarra = 2, Wakayama Medical University Hospital = 4, Osaka City University = 15, Queen Mary Hospital = 5, Fujita Health University Bantane Hospital = 15, Shizuoka Cancer Center = 3, Kitazato University = 1, Komagome Hospital = 4, Koo Foundation Sun Yat-Sen Cancer Center = 2, Far-Eastern Memorial Hospital = 20, Osaka City General Hospital = 32, Kurume University = 1, Kurashiki Central Hospital = 23, National Cancer Center Hospital East = 4, Tulane University = 1, Institute Mutualiste Montsouris = 10, Kansai Rosai Hospital = 33, Tokyo Medical and Dental University = 1, Toho University = 4, Asan Medical Center = 3, Fujita Health University Hospital = 22, and Keio University = 17.

## 4. Discussion

The present study showed that the plasma level of albumin, that of total bilirubin, and ALBI score indicated significantly worsened liver function 3 months after RLR comparing to the preoperative values. Although ALBI [14,15] and ALPlat [16] scores are advocated for liver functional evaluation before HCC treatments including LR, there are no established assessment indicators for the permanent deterioration of liver function settled stable 3 months after treatments. These factors, plasma level of albumin, that of total bilirubin, and ALBI score, could be the candidate indicators for the assessments of liver functional permanent deterioration after LR. Using these indicators, evaluation for the extent of liver functional deterioration after LRLR compared to ORLR were also performed in present study.

With the original patient group for the present study, we conducted international multi-institutional studies for LRLR to HCC patients, compared to ORLR [11,12]. The studies showed that LRLR is feasible and has short-term advantages of less intraoperative blood loss and less morbidity for selected patients. In the first study [11], the overall survival curves after LRLR and ORLR were clearly separated with the better tendency in LRLR, although the disease-free survival curves were identical. Overall survival of HCC patients with CLD after LR is determined not only by the recurrence of the resected HCC, but also by metachronous multicentric HCCs and liver insufficiency [8,9]. During the long and repeated treatment history of patients with HCC, they should have enough residual liver function after each treatment which makes them possible to undergo repeat combination treatments. We hypothesized that overall survival after LRLR was better since less deterioration of liver function after LRLR [12], in addition to less adhesion, made the repeat treatments more accessible and the number of deceased patients due to liver insufficiency decreased. 

The main advantages of LLR for repeat treatments are thought to be less adhesion after LR and less damage to the liver and surrounding structures, such as collateral vessels [17], using the laparoscopic direct approach to the surgical area [18,19,20], sometimes without complete dissection of adhesion. Those could work not only on the technical aspects during LR, but also on the liver function after treatments resulting in less deterioration. Both possible advantages were verified by simple comparison of OO (open repeat LR after open LR) and LL (laparoscopic repeat LR after laparoscopic LR) groups, excluding the patients who underwent both open and laparoscopic procedures, in the present study. Selecting the resections of segmentectomy or less were for minimizing the impact on the deterioration from the decreased functional liver volume. There was no difference in tumor number, size, and location (in anterolateral segments or posterosuperior segments) between OO and LL groups. Thereafter, tumor and surgical factors are similar in both groups compared. On the other hand, the LL group had patients with poorer general (poorer PS and higher BMI) and liver condition (more fibrosis, lower albumin and platelet, worse ALBI/ALPLat/Child–Pugh scores) compared to the OO group. LL group patients with poorer liver and general conditions and similar tumor and surgical factors showed similar deterioration of liver function and resulted in similar overall survival to OO group patients. It could be translated that LL group patients could have gone through repeat LR well, despite the fact that they were allocated to LRLR due to the fear of liver decompensation and morbidity after ORLR. It may show the advantage of LLR, that it could prolong the overall survival of the HCC patients with CLD as a powerful local therapy which can be applied repeatedly with minimal deterioration of liver function.

The deterioration of liver function by each HCC treatment is usually smaller and more difficult to detect than years-long deterioration by CLD, except major hepatectomies which remove a large volume of functional liver. The present study showed that plasma level of albumin, that of total bilirubin, and ALBI score are the possible indicators for the assessment of liver functional permanent deterioration after LR. However, LR should have heaviest damage on liver function among the treatment options and, also, the evaluation of each individual case in different condition should be more difficult. Therefore, further investigations are needed for the assessment of liver functional change after each treatment during repeated treatments for the patients with metachronous multicentric HCCs and CLD.

## Figures and Tables

**Figure 1 cancers-14-02598-f001:**
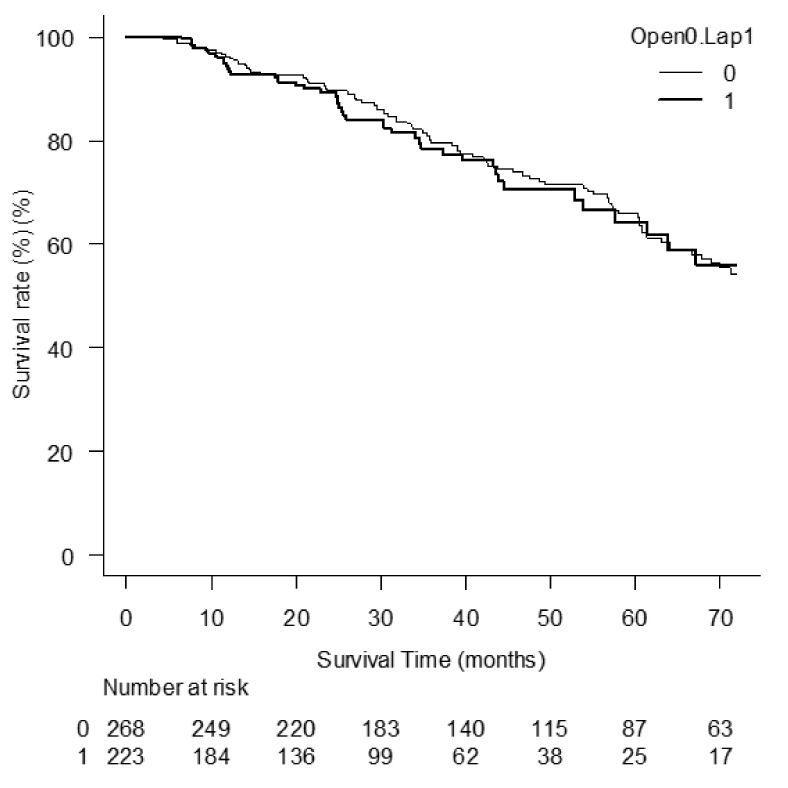
Overall survival after RLR between LL and OO groups.

**Table 1 cancers-14-02598-t001:** Background factors of all patients (*n* = 875) with ORLR or LRLR before RLR.

	ORLR, *n* = 450	LRLR, *n* = 425	*p* Value
**Age (years old)**	66.07 ± 10.77	68.03 ± 10.60	**0.007 ***
**Sex (male:female)**	355:95	322:103	0.270
**BMI**	22.98 ± 3.43	23.98 ± 3.96	**<0.001 ***
**Performance status (0:1:2)**	411:37:1	365:55:5	**0.016 ***
**Size of tumor (mm)**	23.59 ± 17.52	20.49 ± 10.74	**0.002 ***
**Number of tumors (1:2:3:>4)**	315:87:23:25	335:70:13:7	**0.003 ***
**Tumor location (AL:PS)**	159:107	145:77	0.223
**Extent of resection** **(Segment or less: Section: 2 or more sections)**	329:72:49	382:33:10	**<0.001 ***
**Albumin (g/dL)**	4.09 ± 0.41	4.01 ± 0.48	**0.006 ***
**Total Bilirubin (mg/dL)**	0.73 ± 0.32	0.76 ± 0.35	0.095
**Platelet (X10^4^/microL)**	14.77 ± 6.11	13.93 ± 5.10	**0.026 ***
**Presence of fibrosis (NL:CH:LF:LC)**	73:56:114:202 ^#^	49:39:120:21 3 ^##^	0.056
**Child–Pugh score (5:6:7:>8)**	393:46:9:2	322:84:14:5	**<0.001 ***

RLR: repeat liver resection, ORLR: open repeat liver resection, LRLR: laparoscopic repeat liver resection. Data are shown as mean ± SD or number of cases. *: statistically significant. ^#^: There are 5 missing data, ^##^: There are 4 missing data.

**Table 2 cancers-14-02598-t002:** Analysis of the indicators of liver function before and 3 months after repeat liver resection.

	Pre-Operative Data	Post-Operative Data	*p* Value
**Albumin (g/dL)**	4.04 ± 0.45	3.97 ± 0.53	**0.006 ***
**Total Bilirubin (mg/dL)**	0.76 ± 0.33	0.81 ± 0.40	**0.010 ***
**Platelet (×10^4^/microL)**	14.07 ± 5.02	14.12 ± 5.20	0.862
**ALBI score**	−2.73 ± 0.40	−2.65 ± 0.48	**0.001 ***
**AlPlat score**	504.49 ± 70.46	498.24 ± 77.05	0.125

Data are shown as mean ± SD. *: statistically significant.

**Table 3 cancers-14-02598-t003:** Comparison between OO group and LL group: Background factors, indicators for liver function before RLR, and after RLR.

Before LR	OO	LL	*p* Value
**Age (years old)**	67.37 ± 10.36	68.62 ± 9.96	0.176
**Sex (male:female)**	214:54	167:57	0.194
**BMI**	22.94 ± 3.44	23.96 ± 3.98	**0.002 ***
**Performance status (1:2:3)**	250:17:1	194:29:1	**0.043 ***
**Number of tumors (1:2:3:>4)**	188:58:14:8	176:38:6:4	0.209
**Size of tumor (mm)**	20.93 ± 15.21	19.00 ± 9.52	0.089
**Tumor location (AL:PS)**	159:107	145:77	0.223
**Albumin (g/dL)**	4.09 ± 0.39	3.94 ± 0.49	**<0.001 ***
**Total Bilirubin (mg/dL)**	0.73 ± 0.31	0.75 ± 0.35	0.698
**Platelet (×10^4^/microL)**	14.58 ± 4.89	13.57 ± 5.41	**0.031 ***
**ALBI score**	−2.78 ± 0.34	−2.65 ± 0.46	**<0.001 ***
**AlPlat score**	514.32 ± 61.09	490.43 ± 79.95	**<0.001 ***
**Presence of fibrosis (NL:CH:LF:LC)**	48:41:70:106	21:23:63:114	**0.006 ***
**Child-Pugh score (5:6:7:>8)**	239:25:4:0	160:53:7:4	**<0.001 ***
3 months after LR			
**Albumin (g/dL)**	4.03 ± 0.47	3.89 ± 0.55	**0.003 ***
**Total Bilirubin (mg/dL)**	0.77 ± 0.36	0.80 ± 0.39	0.461
**Platelet (X10^4^/microL)**	14.77 ± 5.09	13.68 ± 5.56	**0.025 ***
**ALBI score**	−2.71 ± 0.42	−2.59 ± 0.52	**0.003 ***
**AlPlat score**	510.10 ± 69.08	486.70 ± 83.60	**0.001 ***

Data are shown as mean ± SD or number of cases. *: statistically significant. OO group: Cases who underwent open repeat liver resection after previous open liver resection. LL group: Cases who underwent laparoscopic repeat liver resection after previous laparoscopic liver resection. RLR: repeat liver resection, LR; liver resection, BMI: body mass index, AL: tumors located anterolateral segments (segments 2–6), PS: tumors located posterosuperior segments (segments1,7,8), NL: normal liver, CH:chronic hepatitis, LF: liver fibrosis, LC: liver cirrhosis.

**Table 4 cancers-14-02598-t004:** Comparison between OO group and LL group: Changes in indicators for liver function before and after RLR.

	OO	LL	*p* Value
**Change of Alb (g/dL)**	0.068 ± 0.40	0.054 ± 0.42	0.710
**Change of Total Bilirubin (mg/dL)**	−0.036 ± 0.34	−0.049 ± 0.33	0.653
**Change of Platelet (×10^4^/microL)**	−0.19 ± 4.26	−0.11 ± 3.34	0.830
**Change of ALBI score**	−0.064 ± 0.35	−0.063 ± 0.38	0.969
**Change of ALPlat score**	4.23 ± 53.46	3.73 ± 53.59	0.919

Data are shown as mean ± SD. OO group: Cases who underwent open repeat liver resection after previous open liver resection. LL group: Cases who underwent laparoscopic repeat liver resection after previous laparoscopic liver resection.

## Data Availability

The data presented in this study are available on request from the corresponding author.

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
