# Peer review of "An International Retrospective Observational Study of Liver Functional Deterioration after Repeat Liver Resection for Patients with Hepatocellular Carcinoma"

_cancers, 2022, doi:10.3390/cancers14112598_

Round 1
Reviewer 1 Report
I consider that the revised paper may be published.
Author Response
Dear Reviewer,
Thank you for your kind words.
Your suggestions made our manuscript better and more valuable.
Sincerely,
Zenichi Morise M.D., Ph.D., FACS Professor and Chairman, Department of Surgery Fujita Health University School of Medicine Founding Past Director, Fujita Health University Okazaki Medical Center Deputy Chief Editor, Fujita Medical Journal
Reviewer 2 Report
The manuscript is interesting and overall well written. I just found excessive self-citation by the authors (at least 25% of the references have as first author Morise Z....).
Maybe some other predictors could be analyzed and reported in Table 2.
Author Response
Dear Reviewer,
Thank you for your review comments.
According your suggestion, we removed ref 8 & 16 (self-citations) and put new refs instead of those.
Unfortunately, the data related to liver function 3 months after surgery could not be retrieved sufficiently, due to this study's retrospective nature, other than Alb, TB and Platelet. Many patients were 3 months after surgery in out-patient department or even in the other hospitals than where the surgery underwent, and had done only routine exams.
We tried to retrieve the data, but the retrieved rates were 77, 85, 85, and 81% for prothrombin time, ascites, encephalopathy, and esophageal varices.
Within the analyses of these limited data, there were no significant differences (changes) detected between the values of pre and post LR in neither all patients, OO patients nor LL patients. We did not add this data in the body of manuscript due to their insufficiency.
Thank you again for your suggestions making our manuscript better and more valuable.
Sincerely,
Zenichi Morise M.D., Ph.D., FACS Professor and Chairman, Department of Surgery Fujita Health University School of Medicine Founding Past Director, Fujita Health University Okazaki Medical Center Deputy Chief Editor, Fujita Medical Journal
Round 2
Reviewer 2 Report
The manuscript is OK in the current form
This manuscript is a resubmission of an earlier submission. The following is a list of the peer review reports and author responses from that submission.
Round 1
Reviewer 1 Report
Very well-written review in an important topic.
Comments to improve the manuscripts:
- The authors are suggested to mention a number of cases in each group from every participating center.
- Authors are recommended to present data both befor and after propensity-score matching taking in account preoperative performance status and liver function.
- The abbreviation PS should be explained in the Abstract or it should be written without abbreviation (preoperative performance status). Otherwise, a reader would not understand what PS means in the Abstract.
Reviewer 2 Report
Even though the results of the presented study could be of interest, at present they are rather descriptve than predictive. Iin general, it does not seem surprising that ALBI score etc. is declining early after liver re-resection. However, it could be interesting to correlate early liver function decline with clinical outcome regrading over survival and risk of HCC recurrence. None of these data is presented in the study. In addition, relevamt daat on baseline liver function ( such as Child Pugh score, MELD score) and tumor load (number and size, Milan criteria) prior resection and re-resection has been made available by the authors. Therefore, in its current form clinical usefulness and applicability of the presented data is very limited.
Reviewer 3 Report
This is a well written paper, on a very interesting topic, analyzing, on a large number of patients, the effect of surgical approach (laparoscopic vs open) on the function of impaired background liver after minor (repeated) resections for HCC. The methods are properly designed and applied. The results are valuable, proving the superiority of laparoscopy over open approach in this scenario. I consider that this paper may be published in the current form.